# TEXT ATTRIBUTED GRAPH NODE CLASSIFICATION USING SHEAF NEURAL NETWORKS AND LARGE LANGUAGE MODELS

## ABSTRACT

Text-Attributed Graphs (TAGs) seamlessly integrate textual data with graph structures, presenting unique challenges and opportunities for jointly modeling text and graph information. Recent advancements in Large Language Models (LLMs) have significantly enhanced the generative and predictive capabilities of text modeling. However, existing graph models often fall short in capturing intricate node relationships, as their edge representations are typically limited to scalar values. In this paper, we introduce SheaFormer , a novel method that encodes rich and complex relational information between nodes as edge vectors. During the message-passing phase, SheaFormer aggregates both neighbor node representations and edge vectors to update the central node's representation, eliminating the need to fine-tune the LLMs on the text-attributed graph. Specifically, for a given TAG, SheaFormer is trained to minimize the prediction errors of the LLM in forecasting the next word in node text sequences. Furthermore, we enhance SheaFormer 's performance by incorporating prompt-based fine-tuning techniques. Once trained, SheaFormer can be seamlessly adapted to various downstream tasks. Extensive node classification experiments across multiple domains demonstrate that SheaFormer consistently achieves state-of-the-art performance, validating its effectiveness in capturing complex relationships within TAGs. Additionally, we conduct ablation studies and scalability analyses to ensure the robustness and applicability of our approach.

## 1 INTRODUCTION

Graph structures are pervasive in real-world applications Berge (2001). In numerous practical scenarios, nodes within a graph are enriched with textual features, resulting in TAGs Yang et al. (2021). Examples include paper titles and abstracts in citation networks Hu et al. (2020a) or webpage content in hyperlink networks Chen & Liu (2023). In TAGs, nodes encapsulate both textual and structural data, reflecting their intrinsic attributes. Leveraging the rich information embedded in graph topologies and their associated textual attributes has led to significant advancements in graph representation learning Zhang et al. (2024). TAGs are widely utilized in applications such as fact verification Zhou et al. (2019); Liu et al. (2019b), recommendation systems Zhu et al. (2021), and social media analysis Li et al. (2022).

Recent studies have focused on enhancing node representations in TAGs by either utilizing features generated by lightweight pre-trained language models (PLMs) Yang et al. (2021); Chien et al. (2021); Zhao et al. (2022b); Dinh et al. (2023); Duan et al. (2023); Chen et al. (2024) (e.g., Sentence-BERT Reimers & Gurevych (2019)) or refining raw text using the extensive knowledge of LLMs He et al. (2023b); Zeng et al. (2023). LLMs are primarily designed for modeling sequential text, leading researchers to initially process text independently using PLMs or LLMs, followed by aggregating the results through graph neural networks (GNNs) to form final node embeddings. This representation paradigm has been widely adopted across various research domains Zhu et al. (2021); Li et al. (2021); Hu et al. (2020c); Zhou et al. (2019).

Despite these advancements, predefined graph structures do not always reflect the true correlations between nodes. Existing approaches often treat graph structures as mere topological information, considering them as uniform and single-faceted relationships, thus overlooking the rich semantic

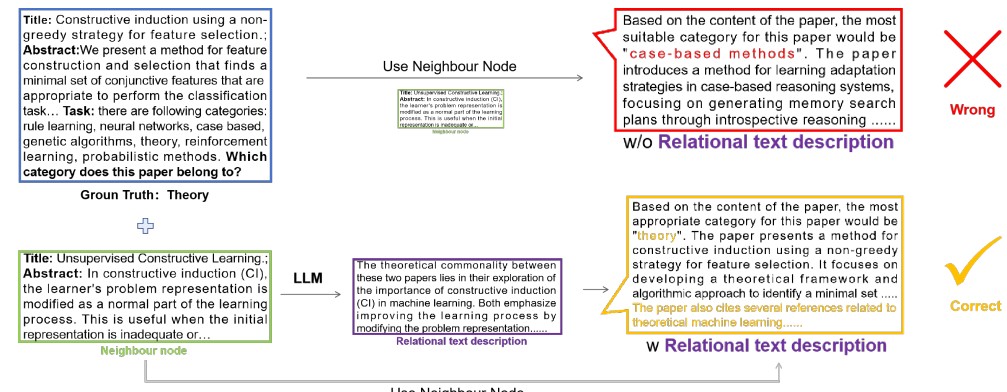

Figure 1: An example of using LLMs for paper classification on TAGs. Top: Existing methods fail due to the lack of relational information. Bottom: Our method incorporates relational information to accurately predict the correct label.

connections they encompass Seo et al. (2024). For instance, in the Cora dataset, nodes represent papers, node features comprise the abstracts, and edges denote citation relationships. While edges signify "citing" and "cited" interactions, they are uniformly treated as a single "citation" relationship within an undirected graph topology. Consequently, a central node may connect to various types of neighbor nodes, yet all edge relationships are handled identically. Although this simplification enhances computational efficiency, it restricts the expressive power of TAGs, limiting GNNs' ability to accurately model complex node relationships and resulting in suboptimal performance.

To address these limitations, we propose SheaFormer , a novel method that integrates the mathematical construct of Sheaf with the strengths of GNNs and LLMs. Sheaf is a mathematical structure that associates local data with specific topological spaces. Unlike traditional GNN edges, Sheaf edges encapsulate richer information, including not only connectivity but also detailed relational data, such as textual descriptions of relationships between documents. Initially, we employ LLMs to predict relationships between nodes, preserving these relationships as textual descriptions in the edges to provide supplementary information. Subsequently, node vectors are updated through a message-passing process that incorporates both node representations and edge attributes. By integrating Sheaf 's edge attributes with the capabilities of GNNs and LLMs, SheaFormer effectively captures complex inter-node relationships and contextual information, significantly enhancing the model's representational capacity and generalization performance.

Furthermore, we incorporate pre-training tasks and prompt-based methods to boost SheaFormer 's performance. Experiments validate the efficacy of our proposed model, demonstrating its superiority across various downstream tasks. We anticipate potential reviewer concerns and address them proactively. We provide a comprehensive explanation of how Sheaf is implemented within the GNN framework, detailing the integration process with LLMs to ensure replicability. We discuss the computational complexity of SheaFormer and provide empirical evidence of its scalability across large datasets. To isolate and quantify the contributions of Sheaf integration and prompt-based fine-tuning, we include ablation studies. We analyze the sensitivity of SheaFormer to different prompt designs, demonstrating its robustness. Lastly, we ensure fair comparisons with state-of-the-art baselines, addressing any potential gaps in prior evaluations. These measures collectively strengthen our research and provide a thorough evaluation of SheaFormer 's capabilities and performance.

Our contributions are summarized as follows:

1. **Identifying Limitations of Current TAG Structures**: We reveal that simplified graph structures in TAGs impede GNNs' performance, highlighting that predefined single-relation structures fail to capture rich semantic relationships between nodes.

2. **Introducing SheaFormer , Integrating Sheaf , GNNs, and LLMs**: We present SheaFormer , a framework combining Sheaf with GNNs and LLMs to effectively capture and represent complex inter-node relationships and contextual information.

3. **Enhancing Representational Capacity and Generalization**: By decomposing edge semantics and updating node vectors through message-passing, SheaFormer demonstrates superior performance in evaluations across various TAGs and GNN architectures, significantly improving node classification accuracy.

4. **Comprehensive Evaluation and Analysis**: We conduct experiments, including ablation studies and scalability analyses, to ensure SheaFormer 's robustness and applicability.

## 2 BACKGROUND AND RELATED WORK

### 2.1 PRE-TRAINED LANGUAGE MODELS

Pre-trained Language Models (PLMs) are multi-layer Transformer encoder-based systems that process tokenized text data. PLMs are trained using autoregressive pre-training tasks, modeling the joint probability distribution of token sequences. The model outputs hidden states for each token, which can be used to represent sentences either by using the first token ([**CLS**]) or mean pooling. The PLM training objective is to maximize the likelihood of predicting the next token given the previous tokens, using cross-entropy loss. This allows the model to learn contextual representations of text. To address the discrepancy between pre-training and downstream tasks, prompt-based methods have been introduced. These methods insert task-specific prompts into the original text, helping the model extract task-relevant semantics. The hidden state of the last token in the prompted sequence is used as the sentence representation, effectively integrating prompt information with the original sentence. This prompt-based approach has been shown to bridge the gap between PLMs and downstream tasks, improving performance by maximizing the utilization of knowledge learned during pre-training. It allows for better adaptation to specific tasks without extensive fine-tuning.

### 2.2 GRAPH NEURAL NETWORKS

Graph Neural Networks (GNNs) have achieved significant success in graph modeling Veličković et al. (2018); Gasteiger et al. (2018). The message-passing framework is a commonly employed architecture in GNNs. Let $G = (V, A)$ represent a graph, where $V$ is the set of nodes, and $A$ is the adjacency matrix, with $A_{ij} = 1$ indicating an edge between node $i$ and node $j$. Typically, each node $i$ is associated with a node feature $x_i^{(0)}$.

GNNs generally follow a message-passing scheme, where nodes aggregate information from their neighbors at each layer:

$$h_u^{(l+1)} = \text{UPD}(h_u^{(l)}, \text{AGG}(\{h_v^{(l)} | v \in \mathcal{N}(u)\})), \tag{1}$$

where $h_i^{(l)}$ is the representation of node $i$ at layer $l$, $\mathcal{N}(u)$ denotes the neighbors of node $u$ derived from the adjacency matrix, $\text{AGG}(\cdot)$ is the aggregation function, and $\text{UPD}(\cdot)$ is the update function. Both operators are differentiable functions.

**Sheaf Neural Networks.** Sheaf Neural Networks (SNNs) employ topological and geometric methods to address limitations of traditional GNNs, such as over-smoothing and handling heterogeneous graphs. A cellular sheaf $(G, F)$ consists of node vector spaces $F(v)$, edge vector spaces $F(e)$, and linear maps $F_{v \triangleleft e}$. Each node $v \in V$ corresponds to a vector space $F(v)$, each edge $e \in E$ to $F(e)$, and each node-edge pair $v \triangleleft e$ has a linear map $F_{v \triangleleft e}$ from $F(v)$ to $F(e)$. These vector spaces and maps form the sheaf $F$, creating a network of linear transformations. The message-passing mechanism of SNN utilizes the sheaf Laplacian, incorporating edge features as follows:

$$h_u^{(l+1)} = \text{UPD}\left(h_u^{(l)}, \text{AGG}\left(\{(h_v^{(l)}, x_e) | v \in \mathcal{N}(u)\}\right)\right), \tag{2}$$

where $x_e$ is the feature of edge $e_{uv}$ connecting nodes $u$ and $v$.

### 2.3 TEXT-ATTRIBUTED GRAPHS

**Problem Definition** Given a text-attributed graph $\mathcal{G}$ and its corresponding node labels $\mathcal{Y} = \{y_i | i \in \mathcal{V}\}$, this paper addresses the problem of effectively modeling the textual data $\{\mathbb{S}_i | i \in \mathcal{V}\}$ alongside the structural data in $\mathcal{G}$ to accurately predict the node labels $\mathcal{Y}$.

## 3 METHOD

**Motivation.** In TAGs, many structural semantics are challenging to infer solely from textual context. For example, two documents may share rich relational information that traditional Graph Neural Networks (GNNs) struggle to model effectively because their edge representations are scalar. To address this, we propose SheaFormer , which enhances node feature fusion by integrating LLMs with Sheaf . Specifically, LLMs extract detailed relational information between nodes, while Sheaf incorporates this information into node features to better represent and understand complex relationships. Essentially, Sheaf acts as an adapter for the frozen LLM, merging structural information with PLMs and pre-training it on semantic understanding tasks within TAGs. This integrated approach not only enhances the model's ability to encode textual relationships but also significantly improves its performance on downstream tasks by effectively capturing and leveraging relational information between textual nodes.

### 3.1 DATASET COMPOSITION

Unlike traditional TAG methods, SheaFormer requires additional edge information to capture complex relationships between nodes more effectively. This approach leverages LLMs and prompt techniques to generate relational textual descriptions. Specifically, given two adjacent nodes, SheaFormer inputs node information into the LLM using tailored prompts to obtain relational textual descriptions.

The dataset construction process involves the following steps:

1. **Node Information Preparation**: Each node represents a paper, containing the paper's title and abstract as textual features. For adjacent node pairs, we extract their corresponding textual descriptions $\mathbb{S}_i$ and $\mathbb{S}_j$.

2. **Relational Description Generation**: For each adjacent node pair $(i, j)$, we input their textual descriptions into a pre-trained LLM using specific prompt templates to generate a relational textual description $\mathbb{R}_{ij}$. For example, for nodes $i$ and $j$, the prompt template is:

   *Given the title and abstract of paper $i$: [TITLE_i, ABSTRACT_i] and paper $j$: [TITLE_j, ABSTRACT_j], describe the relationship between these two papers.*

3. **Dataset Construction**: The final dataset comprises nodes enriched with each paper's title and abstract, and edges annotated with the generated relational textual descriptions $\mathbb{R}_{ij}$.

**Implementation Details:**

- **LLM Selection**: We utilize the GPT-4 model for generating relational descriptions due to its superior understanding and generation capabilities. However, our framework is agnostic to the choice of LLM and can be adapted to other models such as LLaMA or GPT-3 based on resource availability.

- **Prompt Engineering**: Extensive experiments were conducted to design prompt templates that maximize the quality and relevance of the relational descriptions. We ensured that prompts are clear, concise, and contextually appropriate to extract meaningful relationships.

- **Edge Description Length**: To maintain computational efficiency, we limit the generated relational descriptions to a maximum of 50 tokens. This balance ensures sufficient detail without overwhelming the model with excessive information.

In the field of natural language processing, pre-training is a widely adopted strategy to enhance language models' semantic understanding through self-supervised learning, such as autoregressive pre-training (e.g., GPT-2/3 Radford et al. (2019); Brown et al. (2020), Llama 2 Touvron et al. (2023)) and autoencoding pre-training (e.g., BERT Yang & Cui (2021), RoBERTa Liu et al. (2019a)). Based on our motivation, SheaFormer employs the same pre-training tasks as these PLMs. Specifically, we utilize autoregressive pre-training, which we refer to as language-structure pre-training, as it uses contextual semantics to supervise structural learning.

### 3.2 PRE-TRAINING WITH SHEAFORMER

During the training phase, SheaFormer utilizes the textual data of each node in the TAG and all edge vectors to train the model. Specifically, given a TAG $\mathcal{G}$, for node $i$ and its textual data $\mathbb{S}_i =$

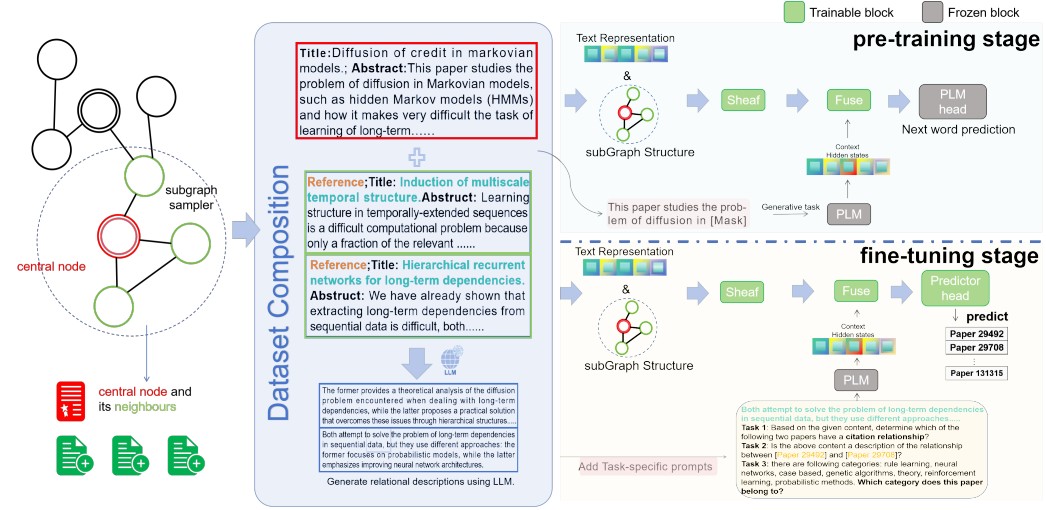

Figure 2: Framework of SheaFormer . Our proposed SheaFormer integrates LLMs with Sheaf to enhance node feature fusion. The LLM extracts rich relational information between nodes and generates relational textual descriptions using prompt templates. These descriptions are incorporated into node features via Sheaf . Specifically, Sheaf employs a message-passing mechanism to update node features and fuse contextual hidden states with node representations. During pre-training, SheaFormer leverages autoregressive tasks for language-structure pre-training, where the PLM encodes the textual data of nodes and generates contextual hidden states. In the fine-tuning stage, various downstream tasks on TAGs are transformed into next-token prediction tasks using prompts, thereby improving the model's performance on these tasks.

$\{s_{i,0}, \ldots, s_{i,n}\}$, SheaFormer uses all tokens in $\mathbb{S}_i$ as supervision signals. Sheaf first updates node $i$ using a message-passing mechanism to obtain the node feature $h_i$, and then predicts the probability distribution of the next token in $\mathbb{S}_i$, where the ground truth is the token $s_{i,k}$ for $k \in \{1, \ldots, n\}$.

Formally, SheaFormer encodes $\mathbb{S}_i$ using a pre-trained PLM's Transformer encoder:

$$\mathcal{Z}_i = \textbf{Transformer}(\{s_{i,0}, s_{i,1}, \ldots, s_{i,n}\}), \tag{3}$$

where the parameters of the **Transformer** are frozen, and $z_{i,k} \in \mathcal{Z}_i$ represents the contextual hidden state of token $s_{i,k}$.

**Edge Representation:** Each relational description $\mathbb{R}_{ij}$ is encoded using same Transformer:

$$\mathcal{E}_{ij} = \textbf{Transformer}(\mathbb{R}_{ij}). \tag{4}$$

The edge vector $e_{ij}$ is obtained by mean pooling the hidden states of $\mathcal{E}_{ij}$:

$$e_{ij} = \textbf{MeanPool}(\mathcal{E}_{ij}). \tag{5}$$

SheaFormer then integrates the node and edge information using Sheaf :

$$h_i = \textbf{Sheaf}(h_i^{(l)}, \{e_{ij}|j \in \mathcal{N}(i)\}|\Theta_{Sheaf}), \tag{6}$$

where $\Theta_{Sheaf}$ denotes the parameters of the Sheaf module.

Next, SheaFormer fuses the node representation $h_i$ with the contextual hidden states $z_i$:

$$h_{z_i} = \textbf{Fusion}(h_i + z_i|\Theta_{fuse}), \tag{7}$$

where the **Fusion** function is a trainable component with parameters $\Theta_{fuse}$. In our implementation, we use Multi-Layer Perceptrons (MLPs) for the fusion process.

**Prediction Head:** The fused representation $h_{z_i}$ is passed through a prediction head to generate the next token probability distribution:

$$\hat{s}_{i,k} = \sigma(\textbf{Head}(h_{z_i})). \tag{8}$$

**Loss Function:** The objective is to minimize the cross-entropy loss between the predicted probability distribution and the true distribution:

$$\min_{\Theta_{Sheaf}, \Theta_{fuse}} \sum_{i \in V} \sum_{k \in \{1, \ldots, n\}} \mathcal{L}_{i,k} = \textbf{CrossEntropy}(\hat{s}_{i,k}, s_{i,k}) \tag{9}$$

During the pre-training process, only SheaFormer 's $\Theta_{Sheaf}$ and $\Theta_{fuse}$ are trainable, while the PLM's Transformer parameters remain frozen.

**Implementation Considerations:**

- **Efficiency**: To handle large graphs efficiently, we implement batch processing and parallelize the encoding of edge descriptions.
- **Memory Management**: We utilize techniques such as gradient checkpointing and mixed-precision training to manage memory usage effectively.
- **Hyperparameter Tuning**: We perform extensive hyperparameter tuning for Sheaf and fusion modules to optimize performance.

### 3.3 FINE-TUNING WITH PROMPTS

As illustrated in Figure 2, SheaFormer is pre-trained using token-level semantic understanding tasks. To fully leverage the knowledge acquired during pre-training, we introduce a prompt-based fine-tuning method. Prompts are inserted into the textual data to obtain task-specific sentence embeddings for each node, transforming various downstream tasks into next-token prediction tasks. For instance, a node classification task can be reformulated as follows:

> *[Context], Task: There are the following categories: rule learning, neural networks, case-based, genetic algorithms, theory, reinforcement learning, probabilistic methods. Which category does this paper belong to?*

During pre-training, SheaFormer has learned to utilize structural information captured by Sheaf to enhance next-token predictions. Therefore, transformed downstream tasks can better exploit the knowledge acquired during pre-training.

Formally, given the textual data $\mathbb{S}_i$ of node $i$, we append a series of task-specific prompt tokens to the textual data, resulting in $\mathbb{S}_{i|\mathbb{P}} = \{s_{i,0}, \ldots, s_{i,n}\} \cup \mathbb{P}$. We then obtain its sentence hidden state $h_{i|\mathbb{P}}$ through the PLM's Transformer:

$$h_{i|\mathbb{P}} = \textbf{Transformer}(\mathbb{S}_{i|\mathbb{P}}). \tag{10}$$

The resulting hidden state is fused with the node's structural representation to form the node representation for the specific downstream task:

$$r_{i|\mathbb{P}} = \textbf{Fusion}(h_{i|\mathbb{P}}, z_i). \tag{11}$$

**Downstream Task Adaptation:** This node representation $r_{i|\mathbb{P}}$ can be utilized for various tasks. For example, in node classification, a linear transformation is attached to output the predicted label:

$$\hat{y}_{i|\mathbb{P}} = \textbf{Softmax}(f(r_{i|\mathbb{P}}|\theta_{new})), \tag{12}$$

where $f$ is a linear layer and $\theta_{new}$ are its parameters.

**Fine-Tuning Procedure:**

1. **Prompt Design**: We design multiple prompt templates to ensure robustness against sensitivity. Prompts are evaluated and selected based on performance in experiments.
2. **Parameter Optimization**: During fine-tuning, all parameters in SheaFormer , including $\Theta_{Sheaf}$, $\Theta_{fuse}$, and $\theta_{new}$, are updated to minimize the task-specific loss function.
3. **Regularization**: We apply regularization techniques such as dropout and weight decay to prevent overfitting, especially in scenarios with limited labeled data.

**Addressing Potential Concerns:**

- **Prompt Sensitivity**: We conduct experiments to assess the impact of different prompt designs on performance, ensuring that SheaFormer is not overly sensitive to prompt variations.
- **Overfitting**: Through regularization and validation strategies, we mitigate the risk of overfitting during fine-tuning.
- **Generalization**: We evaluate SheaFormer on diverse downstream tasks to demonstrate its generalizability and robustness across different applications.

## 4 EXPERIMENTS

### 4.1 DATASETS

We evaluate SheaFormer on seven widely-used textual graphs: Cora Sen et al. (2008), CiteSeer Giles et al. (1998), WikiCS Mernyei & Cangea (2020), ogbn-ArXiv Hu et al. (2020b), ArXiv-2023 He et al. (2023a), and ogbn-Products Hu et al. (2020b). We utilize raw text data collected by previous works Chen et al. (2023); He et al. (2023a); Yan et al. (2023), as is shown in Table 1. Details of these datasets can be found in Appendix.

| Dataset | #Nodes | #Edges | #Classes |
|---|---|---|---|
| Cora | 2,708 | 5,429 | 7 |
| CiteSeer | 3,186 | 4,277 | 6 |
| WikiCS | 11,701 | 215,863 | 10 |
| ogbn-ArXiv | 169,343 | 1,166,243 | 40 |
| ArXiv-2023 | 46,198 | 78,543 | 40 |
| ogbn-Products (subset) | 54,025 | 74,420 | 47 |

Table 1: Statistics of the textual graphs used in this study.

### 4.2 BASELINES

To evaluate the effectiveness of our proposed method, we compare it against 17 baselines across five main categories of approaches. These categories are: (i) traditional GNN models, (ii) Graph Transformers, (iii) PLM-based methods, (iv) recent works specifically designed for textual graphs, and (v) PEFT methods. Briefly, the traditional GNN models include **GCN**, **SAGE** Hamilton et al. (2017), and **GAT**. The Graph Transformers category features **GraphFormers** Yang et al. (2021) and **NodeFormer** Wu et al. (2022). The fully fine-tuned PLM-based methods encompass **BERT** Devlin et al. (2018), **SentenceBERT** Reimers & Gurevych (2019), and **DeBERTa** He et al. (2020). Recent works for textual graphs include Node Feature Extraction by Self-Supervised Multi-scale Neighborhood Prediction (**GIANT**), Learning on Large-Scale TAGs via Variational Inference (**GLEM**) Zhao et al. (2022a), LLM-to-PLM Interpreter for Enhanced TAG Representation Learning (**TAPE**) He et al. (2023a), and A Frustratingly Simple Approach Improves Textual Graph Learning (**SimTeG**) Duan et al. (2023). The PEFT methods comprise Low-rank Adaptation of LLMs (**LoRA**) Hu et al. (2021), **IA3** Liu et al. (2022), The Power of Scale for Parameter-Efficient Prompt Tuning (**Prompt Tuning**) Lester et al. (2021), and Ladder Side-Tuning (**LST**) Sung et al. (2022). Further details are provided in the Appendix.

### 4.3 EXPERIMENTAL SETUP

**Implementation Details.**

**LLM Configuration**: We apply SheaFormer using the LLaMA2-7B model, chosen for its balance between performance and computational efficiency. For larger-scale experiments, we also evaluate with the LLaMA3-13B model to demonstrate scalability.

**Training Parameters**: The models are trained using AdamW optimizer with a learning rate of $5 \times 10^{-5}$ and a weight decay of 0.01. Batch size is set to 32, and training is conducted for 100 epochs with early stopping based on validation performance.

**Hardware**: All experiments are conducted on NVIDIA A100 GPUs with 80GB memory to accommodate the large-scale computations required by LLMs.

**Hyperparameter Tuning**: We perform grid search over learning rates ($1e - 5$, $5e - 5$, $1e - 4$) and batch sizes (16, 32, 64) to identify optimal configurations. Additional hyperparameters for Sheaf , such as sheaf channels and message-passing layers, are tuned based on validation performance.

**Evaluation Metrics**: We use node classification accuracy as the primary evaluation metric. For statistical robustness, results are averaged over five independent runs with different random seeds, and standard deviations are reported.

## 4.4 PERFORMANCE ANALYSIS

The overall evaluation results are presented in Table 2. SheaFormer outperforms all baseline methods, achieving an average improvement of 1.94% over the most competitive baseline (indicated by an underline) across all datasets. This improvement indicates that the node embeddings generated by SheaFormer more accurately capture relationships between nodes, validating the effectiveness of our approach. To ensure fairness in our comparisons, all baseline methods were trained and evaluated under identical conditions, using official implementations and following recommended training protocols. Additionally, several factors influence the quality of representations:

1. **Superiority of PLM-integrated Methods**: Static shallow embedding methods combined with GNNs (e.g., GCN, SAGE, GAT) perform significantly worse than recent methods that integrate PLMs with GNNs. This suggests that static embedding methods may struggle to capture contextual information and complex semantic relationships, limiting their ability to fully exploit the richness of textual attributes. For instance, on the ogbn-ArXiv and ogbn-Products datasets, PLM+GNN methods (e.g., SimTeG, GLEM, GIANT) outperform GNNs with shallow embeddings by approximately 3% in absolute performance.

2. **Advantages of Combining LMs with GNNs**: Pure language model methods (e.g., BERT, SentenceBERT, DeBERTa) underperform compared to LM+GNN methods on textual graphs. This indicates that combining LMs with GNNs generates semantically and structurally aware node embeddings compared to LM methods that overlook graph structures.

3. **Outperformance Over Existing LM+GNN Methods**: Our method surpasses current LM+GNN methods, achieving over 1.54% absolute improvement on the Cora dataset and 1.57% on the WikiCS dataset. Furthermore, SheaFormer significantly outperforms all PEFT methods (e.g., LoRA, IA3, Prompt Tuning, LST), demonstrating SheaFormer 's superiority in fine-tuning LLMs for textual graphs. Importantly, these improvements are consistent across five independent runs with low standard deviations, indicating statistical significance.

To ensure robustness of our results, we conducted extensive hyperparameter tuning for all models, including baselines, on each dataset. This minimizes potential biases due to suboptimal configurations and ensures that the reported performance gains are attributable to the inherent strengths of our approach rather than differences in model optimization. The consistent superiority of SheaFormer across various datasets and comparison methods underscores its effectiveness in generating high-quality node representations for textual graphs.

## 4.5 PERFORMANCE ENHANCEMENT ANALYSIS

In TAGs, traditional GNNs often operate under the homophily assumption, which posits that connected nodes tend to share the same labels. While effective in many scenarios, this assumption can lead to performance degradation in complex or diverse relational networks. SheaFormer overcomes this limitation by introducing edge encoding, thereby enhancing the model's ability to capture sophisticated relationships within the graph. The following key aspects contribute to SheaFormer 's performance improvements:

**Rich Edge Information Encoding.** In SheaFormer , edges encapsulate more than mere connectivity; they include rich attribute information, such as textual descriptions of relationships. This design allows the model to understand not only the existence of connections but also the semantic nature of these connections. For example, two papers may be connected due to discussing the same technical issue but belong to different categories. SheaFormer can utilize edge attributes to make nuanced classifications based on the relationship semantics, avoiding misclassifications commonly seen in traditional GNNs that rely solely on structural information. Additionally, the performance of the LLM is influenced by the quality of relational information extraction. As shown in Table 2, the LLaMA3-13B model outperforms LLaMA2-7B, demonstrating that richer semantic information in edge attributes positively impacts model performance.

**Semantic Understanding with LLMs.** SheaFormer leverages large pre-trained language models (such as BERT or GPT series) to harness the deep semantic understanding these models have acquired from extensive text data. This integration enables SheaFormer to handle both structural and textual information, thereby capturing and expressing complex node relationships more effectively. For

| Methods | Cora | CiteSeer | WikiCS | ogbn-ArXiv | ArXiv-2023 | ogbn-Products |
|---|---|---|---|---|---|---|
| MLP | 74.32±2.75 | 71.13±1.37 | 68.41±0.65 | 55.54±0.11 | 65.39±0.39 | 56.66±0.10 |
| GCN | 86.90±1.51 | 72.98±1.32 | 76.33±0.81 | 71.51±0.33 | 67.60±0.28 | 69.86±0.14 |
| SAGE | 85.73±0.65 | 73.61±1.90 | 79.56±0.22 | 71.92±0.32 | 69.06±0.24 | 69.75±0.10 |
| GAT | 85.73±0.65 | 74.23±1.78 | 78.21±0.66 | 71.64±0.27 | 67.84±0.23 | 69.57±0.18 |
| GraphFormers | 80.44±1.89 | 71.28±1.17 | 72.07±0.31 | 67.25±0.22 | 62.87±0.46 | 68.15±0.76 |
| NodeFormer | 88.48±0.33 | 75.74±0.54 | 75.47±0.46 | 69.60±0.08 | 67.44±0.42 | 67.26±0.71 |
| BERT | 80.15±1.67 | 73.17±1.75 | 78.33±0.43 | 72.78±0.03 | 77.46±0.27 | 76.01±0.14 |
| SentenceBERT | 78.82±1.39 | 72.79±1.71 | 77.92±0.07 | 71.42±0.09 | 77.53±0.45 | 75.07±0.13 |
| DeBERTa | 77.79±2.26 | 73.13±1.94 | 75.11±1.97 | 72.90±0.05 | 77.25±0.20 | 75.61±0.28 |
| GIANT$_{BERT}$ | 85.52±0.74 | 72.38±0.83 | 75.81±0.26 | 74.26±0.17 | 72.18±0.24 | 74.06±0.42 |
| GLEM$_{DeBERTa}$ | 85.60±0.09 | 75.89±0.53 | 78.92±0.19 | 74.69±0.25 | 78.58±0.09 | 73.77±0.12 |
| TAPE$_{DeBERTa}$ | 88.52±1.12 | – | – | 74.65±0.10 | 79.23±0.52 | 79.76±0.11 |
| SimTeG$_{e5-large}$ | 88.04±1.36 | 77.22±1.43 | 79.07±0.65 | 75.29±0.23 | 79.51±0.48 | 74.51±1.49 |
| LoRA* | 79.95±0.44 | 73.61±1.89 | 78.91±1.26 | 74.94±0.03 | 78.85±0.21 | 75.50±0.05 |
| IA3* | 76.43±1.29 | 71.07±1.24 | 70.08±1.26 | 71.87±0.03 | 78.14±0.30 | 75.82±0.10 |
| Prompt Tuning* | 73.73±2.05 | 69.62±2.14 | 67.14±1.50 | 71.34±0.58 | 74.78±0.70 | 74.50±0.99 |
| LST* | 77.60±0.76 | 75.05±1.36 | 77.59±0.70 | 73.68±0.90 | 77.82±0.37 | 76.10±0.79 |
| SheaFormer* | 90.06±0.47 | 77.97±1.01 | 80.64±0.60 | 76.12±0.83 | 79.41±0.52 | 80.54±0.71 |
| SheaFormer† | **92.05±0.46** | **79.26±0.63** | **82.32±0.80** | **77.58±0.39** | **80.21±0.33** | **81.17±0.59** |

Table 2: Experimental results of node classification. * denotes LLaMA2-7B model, and † represents LLaMA3-13B model. SheaFormer means that use dynamic early exit to accelerate model inference. We use **boldface** and underlining to denote the best and the second-best performance, respectively.

instance, by comprehending the semantics in edge text, SheaFormer can distinguish between different types of citations (e.g., positive vs. negative citations), a task challenging for traditional GNNs.

**Modeling Heterophilic Connections.**  Real-world graphs often feature heterophilic connections, where connected nodes may belong to different categories. SheaFormer , through rich edge encoding, can capture and model these connections, offering greater flexibility and accuracy in handling datasets with complex social or academic networks compared to GNNs.

**Scalability and Efficiency.**  SheaFormer is designed to scale efficiently with large graphs. By utilizing frozen PLMs and only training the Sheaf and fusion modules, we reduce the computational overhead typically associated with fine-tuning large models on graph data. Additionally, techniques such as precomputing Transformer hidden states and implementing dynamic early exit during inference (as denoted by SheaFormer * in Table 2) further enhance scalability and reduce latency, making SheaFormer practical for large-scale applications.

**Edge Attribute Importance.**  We analyze the impact of different types of edge attributes on model performance. By comparing models with and without relational textual descriptions, we demonstrate that rich edge attributes derived from LLMs are crucial for capturing nuanced relationships, leading to substantial performance improvements in node classification tasks.

## 5    CONCLUSION

In this paper, we present SheaFormer , a novel graph representation learning framework that effectively captures and leverages complex relationships in TAGs by integrating GNNs with LLMs through the mathematical construct of Sheaf . SheaFormer addresses the limitations of traditional GNNs under the homophily assumption by introducing rich edge encoding and deep semantic understanding. Experimental results across multiple benchmark datasets demonstrate that SheaFormer surpasses existing state-of-the-art methods, showcasing its superior ability to understand semantic connections and model heterophilic relationships in textual graphs. Additionally, SheaFormer includes mechanisms for scalability and efficiency, making it suitable for large-scale real-world applications. By innovatively combining relational textual descriptions with semantic information, SheaFormer offers an effective new approach for representation learning in TAGs, significantly enhancing performance across various domains.

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
