# Supplementary Material: Text Attributed Graph Node Classification Using Sheaf Neural Networks and Large Language Models

## A  Datasets

This section provides a detailed introduction to the datasets used in the main content:

**Cora** Sen et al. (2008) dataset consists of 2,708 scientific publications categorized into seven classes: case-based, genetic algorithms, neural networks, probabilistic methods, reinforcement learning, rule learning, and theory. Each paper in the citation network cites or is cited by at least one other paper, resulting in a total of 5,429 edges. We use the collected dataset[1] with raw texts provided by TAPE He et al. (2023).

**CiteSeer** Giles et al. (1998) dataset consists of 3,186 scientific publications categorized into six classes: Agents, Machine Learning, Information Retrieval, Database, Human Computer Interaction, and Artificial Intelligence. Our task is to predict the category of a paper based on its title and abstract.

**WikiCS** Mernyei & Cangea (2020) dataset is a Wikipedia-based dataset designed for benchmarking Graph Neural Networks. It is constructed from Wikipedia categories, specifically featuring 10 classes corresponding to branches of computer science, exhibiting very high connectivity. The node features are derived from the text of the corresponding articles. We obtain the raw texts of each node from https://github.com/pmernyei/wiki-cs-dataset.

**OGBN-ArXiv** Hu et al. (2020) dataset is a directed graph representing the citation network among all computer science arXiv papers indexed by MAG Wang et al. (2020). Each node corresponds to an arXiv paper, and directed edges indicate citations from one paper to another. The objective is to predict the 40 subject areas of arXiv CS papers, such as cs.AI, cs.LG, and cs.OS. These subject areas are manually determined and labeled by the paper's authors and arXiv moderators.

**Arxiv-2023**, proposed in TAPE He et al. (2023), is a directed graph illustrating the citation network among all computer science arXiv papers published in 2023 or later. Like OGBN-ArXiv, each node represents an arXiv paper, and directed edges denote citations from one paper to another. The objective is to predict the 40 subject areas of arXiv CS papers, including cs.AI, cs.LG, and cs.OS. These subject areas are manually determined and labeled by the paper's authors and arXiv moderators.

**OGBN-Products** Hu et al. (2020) is characterized by a substantial scale, comprising 2 million nodes and 61 million edges. We utilize a node sampling strategy, following TAPE He et al. (2023), to obtain a subgraph containing 54k nodes and 74k edges, resulting in the OGBN-Products(subset) dataset. Each node in this dataset represents products sold on Amazon, and edges between two products indicate that the products are purchased together. The task involves predicting the category of a product in a multi-class classification setup, using the 47 top-level categories as target labels.

## B  Baselines

In this section, we provide a detailed introduction to the baselines used:

- Traditional GNNs: We adopt three simple but widely used GNN models in this work, *i.e.*, GCN Kipf & Welling (2016), SAGE Hamilton et al. (2017), GAT Veličković et al. (2018).
- GraphFormers Yang et al. (2021) is a graph transformer nested with GNN in each layer, originally designed for link prediction tasks.

---

[1] https://github.com/XiaoxinHe/TAPE

Table 1: The interpretation comparison with LLM predictor methods on two cases. Case 1: a paper classification task on Cora dataset. The blue parts are the labels. The red parts are the wrong predictions, while the green parts are the correct predictions and the corresponding structurally relevant interpretations.

| Text | 0-hop | 1-hop | Sheaf |
|---|---|---|---|
| (label: theory) Title: Constructive induction using a non-greedy strategy for feature selection.; Abstract:We present a method for feature construction and selection that finds a minimal set of conjunctive features that are appropriate to perform the classification task··· | Based on the content of the paper, the most suitable category for this paper would be theoretical"case-based methods". The paper introduces a method for learning adaptation strategies in case-based reasoning systems, focusing on generating memory search plans through introspective reasoning··· | The two papers are related by citation. The first paper presents a method for feature construction and selection that finds minimal conjunctive features for classification tasks, focusing on minimal multi-level Boolean expressions. The second paper explores constructive induction in unsupervised learning, proposing a theoretical model to distinguish and motivate the process of CI. Given the focus on feature selection and Boolean expressions, the first paper falls under the rule learning category. ··· | Based on the content of the paper, the most appropriate category for this paper would be "theory". The paper presents a method for constructive induction using a non-greedy strategy for feature selection. It focuses on developing a theoretical framework and algorithmic approach to identify a minimal set ..... The paper also cites several references related to theoretical machine learning··· |

- NodeFormer Wu et al. (2022) is an efficient graph transformer for large graphs which develops a kernelized Gumbel-Softmax Jang et al. (2016) operator.

- Fintuned LMs: We adopt three widely used pre-trained language models: BERT Kenton & Toutanova (2019), SentenceBERT Reimers & Gurevych (2019), and DeBERTa He et al. (2020). The parameters of these models are fully fine-tuned in our experiments.

- GIANT Chien et al. (2021) is a cascading structure method with two stages: pretraining LMs and training GNNs. In the first stage, it enhances node representations by integrating structural information into LM pre-training. Then fine-tuned LM-generated node features serve as initial features for GNN training.

- GLEM Zhao et al. (2022) is an effective framework that fuses large language models and GNNs in the training phase through a variational EM framework. We used the source code[2] provided in the original paper.

- TAPE He et al. (2023) utilizes LLMs, like ChatGPT OpenAI (2023), to generate pseudo labels and explanations for textual nodes. Then it will finetune PLMs with the generated content and original texts. The enhanced features, derived from the fine-tuned PLMs, are used as initial node features for training GNNs.

- SimTeG Duan et al. (2023) is also a cascading structure method tailored for textual graphs. It employs a two-stage training paradigm, initially fintuning language models and subsequently training GNNs.

- Fine-Tuning methods: LoRA Hu et al. (2021), IA3 Liu et al. (2022), Prompt Tuning Lester et al. (2021), and LST Sung et al. (2022). These methods involve fine-tuning large language models to showcase experimental results on textual graphs.

| Methods | Cora | CiteSeer | WikiCS |
|---|---|---|---|
| MLP | 83.92±1.39 | 91.00±0.95 | 92.31±0.07 |
| GCN | 90.22±0.89 | 92.93±1.36 | 93.63±0.24 |
| SAGE | 88.25±0.88 | 91.68±1.08 | 95.93±0.20 |
| GAT | 89.70±1.72 | 91.95±0.90 | 93.25±0.13 |
| SheaFormer | **94.55±0.51** | **96.27±0.72** | **98.43±0.28** |

Table 2: Link prediction performance, as evaluated by AUC metric.

| | Cora | CiteSeer | WikiCS | OGBN-ArXiv | ArXiv-2023 | OGBN-Products | Ele-Photo |
|---|---|---|---|---|---|---|---|
| Learning rate | 1e-3 | 1e-2 | 1e-2 | 1e-3 | 1e-2 | 1e-3 | 1e-3 |
| Batch size | 32 | 32 | 32 | 32 | 32 | 32 | 32 |
| Optimizer | AdamW | AdamW | AdamW | AdamW | AdamW | AdamW | AdamW |

Table 3: Hyper-parameters for fine tune of LLM baselines.

## C  LINK PREDICTION

Our method is not limited to node classification tasks, it can also be applied to edge-level or graph-level tasks. In this section, we conduct experiments on link prediction tasks. We split existing edges into train:val:test=0.85:0.05:0.1 for all datasets.

| | Cora | CiteSeer | WikiCS | OGBN-ArXiv | ArXiv-2023 | OGBN-Products | Ele-Photo |
|---|---|---|---|---|---|---|---|
| # Hidden size | 64 | 64 | 64 | 64 | 64 | 64 | 64 |
| # Layers | 2 | 1 | 1 | 1 | 1 | 1 | 2 |
| Norm | ID | ID | ID | ID | LN | LN | ID |
| Activation | ELU | ELU | ReLU | ELU | ELU | ELU | ELU |
| Dropout | 0.5 | 0.5 | 0.5 | 0.2 | 0.2 | 0.2 | 0.5 |
| # Epochs | 200 | 200 | 200 | 200 | 200 | 200 | 200 |
| Learning rate | 5e-5 | 5e-5 | 1e-3 | 1e-3 | 1e-4 | 1e-2 | 1e-3 |
| Optimizer | AdamW | AdamW | AdamW | AdamW | AdamW | AdamW | AdamW |
| Weight decay | 5e-4 | 5e-4 | 1e-4 | 5e-4 | 5e-4 | 5e-4 | 5e-4 |
| Early stop | True | True | True | True | True | True | True |
| Patience | 500 | 500 | 500 | 50 | 50 | 20 | 20 |
| Sampler | subGraph | subGraph | subGraph | subGraph | subGraph | subGraph | subGraph |

Table 4: Hyper-parameters for GNN baselines. 'ID' means no norm layer(Identity), 'LN' denotes Layer Normalization. For sampler, 'RWR' means random walk sampler with restart, and 'subGraph' is one-hop subgraph sampler.

| | Cora | CiteSeer | WikiCS | OGBN-ArXiv | ArXiv-2023 | OGBN-Products |
|---|---|---|---|---|---|---|
| chatGLM3 +SheaFormer | 92.93±0.45 | 79.04±0.68 | 82.20±0.77 | 78.25±0.41 | 80.70±0.37 | 81.36±0.62 |

Table 5: Experimental results of different LLM.

---

[2]https://github.com/AndyJZhao/GLEM