# OpenReview forum: "Text Attributed Graph Node Classification Using Sheaf Neural Networks and Large Language Models"
_ICLR.cc/2025/Conference — Submitted to ICLR 2025_

### Official Review · Reviewer_ucvj · 2024-10-19

**Soundness:** 1
**Presentation:** 2
**Contribution:** 2
**Rating:** 1
**Confidence:** 4

**Summary:**

The paper presents SheaFormer, a novel framework for Text-Attributed Graphs that integrates Sheaf Neural Networks with Large Language Models to capture complex node relationships. Unlike traditional GNNs, SheaFormer encodes relational information as edge vectors and updates node representations without needing to fine-tune LLMs. The results show that it achieves state-of-the-art performance in node classification tasks.

**Strengths:**

To my knowledge, this is the first work that combines Sheaf Neural Networks with LLMs and applies them to TAGs, providing significant inspiration for the community.

**Weaknesses:**

There is a lack of sufficient experiments to support the authors' claims. First, the authors only conducted node classification experiments. To comprehensively evaluate the performance of the proposed SheaFormer, graph classification and link prediction experiments should be added. While there are some link prediction results in the appendix, the authors do not compare them with most baseline methods. Second, there is a lack of necessary ablation studies to demonstrate the effectiveness of the proposed model components. Third, the authors use LLM to generate relational text between nodes, while the baseline methods do not utilize this information. Is this comparison fair?

**Questions:**

- The presentation of Figure 1 is not very effective, as some of the text is too small to read clearly.
- For large graph datasets, the authors' proposed method requires the additional use of a large model to generate relational text. Could this lead to excessive costs?
- Since the authors' proposed method requires additional information about the edges in the dataset, I would like to know the details of the link prediction experiments conducted by the authors.

---

### Official Review · Reviewer_LQYx · 2024-10-26

**Soundness:** 2
**Presentation:** 2
**Contribution:** 2
**Rating:** 5
**Confidence:** 4

**Summary:**

This paper proposed a new framework for text attributed graph node classification based on a combination of Sheaf Neural Network (SNN), Large Language Model (LLM) and Pretrained Language Model (PLM).  The proposed approach first let LLM generate rich text description of edge. And use graph and text features to pretrain the SNN. After pretraining, a fine-tune stage is performed to evaluate the performance of downstream task. The experimental results shows that the proposed outperforms simple GNNs, simple PLMs and strong GNN + PLM baseline.

**Strengths:**

1. The paper proposed a novel framework that combines LLM and Sheaf Neural Network
2. The performance of the proposed approach is good.

**Weaknesses:**

1. The motivation behind using SNN is unclear. In the paper the author claim that `Sheaf is a mathematical structure that
associates local data with specific topological spaces. Unlike traditional GNN edges, Sheaf edges
encapsulate richer information, including not only connectivity but also detailed relational data, such
as textual descriptions of relationships between documents`. However,  there can be more easier ways to incorporate edge information into final representation of node, such as simply concat the edge representation to the node feature and apply any GNN.


2. The contribution of different parts are unclear and lack of ablation study. In the paper, the author proposed  `a. SNN to incorporate edge information`, `b. LLM generated information to enrich the edge representation` and `c. Use a pretrained-finetune paradigm to build the model`. But it remains unknown what's the contribution of each parts.


3. Unknown baseline settings. The author mentioned multiple baselines in paper, and they can be categorized into `a. simple GNN`, `b. simple PLM`, `c. PLM + GNN`, `d. finetuned LLM`. But there is no mention that what is the input of those data. And whether the feature used in SheaFormer is used in baselines. This makes experimental results questionable.


4. Some important baselines are not included.
For simple PLM models:
- `SPECTER: Document-level Representation Learning using Citation-informed Transformer`
- `LinkBERT: Pretraining Language Models with Document Links`

  For Graph + PLM pretrained model:
- `Patton: Language Model Pretraining on Text-rich Networks`
- `GLaM: Fine-Tuning Large Language Models for Domain Knowledge Graph Alignment via Neighborhood Partitioning and Generative Subgraph Encoding`
- `WalkLM: A Uniform Language Model Fine-tuning Framework for Attributed Graph Embedding`
- No specific paper, but any PLM + any GNN baseline.
- `Sheaf` only baseline
- any GNN (including GCN, GAT etc..) with concatenated edge feature instead of Sheaf.

5. Figure 1 is hard to follow and not intuitive, maybe use a better and more vivid example.

**Questions:**

1. What's the motivation of using SNN ?

2. What's the contribution of SNN, LLM edge feature generation, and pre-training ? Is there any ablation study ?

3. What are the features used in baselines. Are the same set of feature used ?

---

### Official Review · Reviewer_woEC · 2024-10-31

**Soundness:** 2
**Presentation:** 2
**Contribution:** 2
**Rating:** 3
**Confidence:** 4

**Summary:**

The paper presents SheaFormer, a novel method for text-attributed graph (TAG) node classification. The proposed method integrates Large Language Models (LLMs) with Sheaf Neural Networks and Graph Neural Networks (GNNs) to capture complex relationships within TAGs. The paper provides a comprehensive evaluation of SheaFormer's performance on multiple datasets and compares it to various baseline methods. The results show that SheaFormer consistently outperforms existing methods, demonstrating its effectiveness in capturing rich semantic relationships within TAGs.

**Strengths:**

1. **Identifying Limitations of Current TAG Structures**: The paper effectively reveals that simplified graph structures in TAGs hinder GNN performance, emphasizing that predefined single-relation structures fail to capture the rich semantic relationships between nodes.
2. **Extensive Experimental Validation**: The experiments conducted validate the effectiveness of the proposed framework, demonstrating its capabilities across various scenarios.

**Weaknesses:**

1. **Limited Novelty**: Some existing work has already identified the importance of edges, as seen in [1] and [2]. Additionally, many edge-aware GNN models have adopted operations similar to Sheaf Neural Networks, such as [4]. Furthermore, the approach of fusing GNN embeddings with text embeddings to predict the next word has been previously utilized in [5].
2. **Domain Limitation of Datasets**: Expanding the evaluation to include more diverse domains, such as those available in the TEG-DB datasets [1], which feature rich node and edge text, would strengthen the findings.
3. **Narrow Applicability**: The model’s applicability is somewhat restricted to specific tasks within graph domains, such as TAG node classification. The authors should consider its potential for other important tasks, like link prediction.


[1] "TEG-DB: A Comprehensive Dataset and Benchmark of Textual-Edge Graphs." NeurIPS 2024.

[2] "Edgeformers: Graph-empowered transformers for representation learning on textual-edge networks." ICLR 2023.

[3] "Design space for graph neural networks." NeurIPS 2020.

[4] "Can we soft prompt LLMs for graph learning tasks?." Companion Proceedings of the ACM on Web Conference 2024.

**Questions:**

Please see the Weaknesses above.

---

### Official Review · Reviewer_ZMmY · 2024-11-04

**Soundness:** 2
**Presentation:** 3
**Contribution:** 1
**Rating:** 3
**Confidence:** 5

**Summary:**

This paper presents SheaFormer, a new method for Text-Attributed Graphs (TAGs) that encodes complex node relationships as edge vectors, improving on traditional graph models. SheaFormer updates node representations by integrating edge vectors and neighboring nodes without fine-tuning Large Language Models (LLMs) on TAGs. The approach achieves state-of-the-art performance across domains, effectively capturing intricate relationships in TAGs, with robustness and scalability verified through extensive studies.

**Strengths:**

1. This paper is well-written

**Weaknesses:**

1. **Lack of novelty.** In the *PRE-TRAINING WITH SHEAFORMER* phase, Equations (3)–(9) in this paper closely resemble Equations (8)–(11) from "Can GNN be Good Adapter for LLMs?" [1] in both core ideas and equation style, with the only difference being the addition of relational representations. Furthermore, the overall framework—TAG pretraining followed by add task-specific prompt for downstream inference—also aligns with [1].

2. **Missing critical baseline:** The paper lacks comparison experiments with GraphAdapter [1].

3. **Lack of extensive ablation studies:** Specifically, (1) how the model performs without pretraining, (2) the impact of different prompts on results, (3) compatibility with different language models, and (4) whether using edge information improves performance.

4. **Lack of many crucial citations:** The paper lacks citations to [1], [2], [3], [4], [5]...etc., as well as comparisons with these works.

**Reference**


[1] Xuanwen Huang, Kaiqiao Han, Yang Yang, Dezheng Bao, Quanjin Tao, Ziwei Chai, and Qi Zhu. "Can GNN be Good Adapter for LLMs?." In Proceedings of the ACM on Web Conference 2024, pp. 893-904. 2024.

[2] Jiabin Tang, Yuhao Yang, Wei Wei, Lei Shi, Lixin Su, Suqi Cheng, Dawei Yin, and Chao Huang. "Graphgpt: Graph instruction tuning for large language models." In Proceedings of the 47th International ACM SIGIR Conference on Research and Development in Information Retrieval, pp. 491-500. 2024.

[3] Ziwei Chai, Tianjie Zhang, Liang Wu, Kaiqiao Han, Xiaohai Hu, Xuanwen Huang, and Yang Yang. "Graphllm: Boosting graph reasoning ability of large language model." arXiv preprint arXiv:2310.05845 (2023).

[4] Zirui Guo, Lianghao Xia, Yanhua Yu, Yuling Wang, Zixuan Yang, Wei Wei, Liang Pang, Tat-Seng Chua, and Chao Huang. "Graphedit: Large language models for graph structure learning." arXiv preprint arXiv:2402.15183 (2024).

[5] Yuhan Li, Peisong Wang, Xiao Zhu, Aochuan Chen, Haiyun Jiang, Deng Cai, Victor Wai Kin Chan, and Jia Li. "GLBench: A Comprehensive Benchmark for Graph with Large Language Models." arXiv preprint arXiv:2407.07457 (2024)

**Questions:**

See Weaknesses

---

### Meta-Review · Area_Chair_P8DQ · 2024-12-07

**Metareview:**

The paper introduces SheaFormer, a method that enhances Text-Attributed Graphs (TAGs) by encoding complex node relationships as edge vectors. This approach integrates Large Language Models (LLMs) with Sheaf Neural Networks to capture rich semantic relationships, achieving state-of-the-art performance in node classification tasks across multiple datasets. However, the paper’s novelty is somewhat limited, with similar approaches previously explored. Additionally, it lacks critical baselines and sufficient ablation studies, and its applicability is somewhat restricted to specific graph-related tasks. The evaluation could also benefit from more diverse datasets and additional tasks like link prediction.

**Additional Comments On Reviewer Discussion:**

No rebuttal was provided by the authors. No discussion was happened.

---

### Decision · Program_Chairs · 2025-01-22

Reject